# Chronic kidney disease among HIV-positive Zambian adults with tenofovir-associated nephrotoxicity at University Teaching Hospital (UTH) in Lusaka

Edgar Muchinta [ID][1][☉]*, Freeman W. Chabala[2][☉]

1 Laboratory Department, National Heart Hospital, Lusaka, Zambia, 2 Infectious Diseases Department, Levy Mwanawasa Medical University, Lusaka, Zambia

☉ These authors contributed equally to this work.
* emuchinta@gmail.com

## Abstract

Tenofovir disoproxil fumarate (TDF) is a nucleotide reverse transcriptase inhibitor (NRTI) widely used in first-line antiretroviral therapy (ART). Despite its efficacy, TDF has been associated with nephrotoxicity, particularly in patients with renal impairment. It is with this background that most countries including Zambia are replacing TDF-based regimens with Tenofovir Alafenamide (TAF). This study aimed to determine the association between TDF-induced nephrotoxicity at three months and chronic kidney disease (CKD) within five years among people living with HIV (PLWH) in Zambia. A retrospective cohort study of 182 PLWH was conducted at the Adult Center for Infectious Disease Research (AIDC) in Lusaka, Zambia. The incidence of CKD and factors associated with its development in PLWH who initiated TDF were evaluated. Kidney function trends were monitored over five years. Statistical analysis, including the Mixed-Effect model, and the Cox Proportional Hazards Regression model, were conducted to assess the relationship between early nephrotoxicity and long-term CKD. A total of 63 (34.62%) out of 182 files showed that the patients developed CKD, contributing to a total person-time of 910 person-years. The incidence rate of CKD was 69.2 cases per 1000 person-years. The findings indicated a significant association between TDF-associated nephrotoxicity and the development of CKD within five years. The mixed-effects model accounted for population-level trends and individual variability, ensuring robust results. Findings showed that removing outliers made the model more representative, with a significant decline in kidney function over time. The Cox regression model identified male sex and baseline creatinine as substantial risk factors for CKD, with good model fit and discriminatory power. Given the findings, it is recommended that regular monitoring and early intervention strategies be reinforced for patients. Furthermore, continuous evaluation of kidney function

**Data availability statement:** All relevant data are within the manuscript and its Supporting Information files.

**Funding:** The author(s) received no specific funding for this work.

**Competing interests:** The authors have declared that no competing interests exist.

over time is crucial to effectively manage and mitigate the risks associated with CKD in PLWH.

## Introduction

In 2021, it was estimated that over 38.4 million people were living with human immunodeficiency virus (HIV), globally and that about 1.5 million people became newly infected in that year [1]. Over 650 000 people died from acquired immune deficiency syndrome (AIDS)-related illnesses in 2021 [2]. In Africa alone, over 25 million people were living with HIV as reported by World Health Organisation (WHO) African Region in 2018 [3]. In Zambia, approximately 1.3 million people had HIV in 2021 with an overall prevalence of about 11.0% of adults aged 15+ years [4]. HIV prevalence was 13.9% among women and 8.0% among men [4].

Antiretroviral (ARV) medications are widely available and used for the management of persons living with HIV (PLWH). However, there have been concerns about the safety of these drugs [5]. The nucleotide reverse transcriptase inhibitor (NRTI) tenofovir disoproxil fumarate (TDF) is the frequently used ARV as the first-line drug combination [6]. Nephrotoxicity with tubular dysfunction, with or without a decrease in glomerular filtration rate, is the predominant adverse reaction to TDF usage [7,8]. Since TDF is primarily eliminated in an unaltered form via the kidneys [9], patients with moderate or severe renal impairment have significantly lower TDF clearance. In such patients, TDF use is preferentially avoided [10]. Previous observational studies have produced mixed findings about the possibility of progressive kidney impairment brought on by long-term TDF exposure [11]. Even in patients with normal renal function before the introduction of TDF, certain studies have shown a correlation between cumulative exposure to TDF and the risk of chronic kidney disease (CKD) [10,12,13]. However, other studies have indicated that the risk of progression is comparatively low over the long term and that decrease in kidney function occurs primarily during the first year of TDF exposure [10,14,15]. The results of a study conducted in Zambia introduced a predictive model that could only predict nephrotoxicity up to the first three months of TDF use and could not tell whether the nephrotoxicity led to CKD later on or not [8].

One of the most significant non-communicable co-morbidities among PLWH, both in developed and resource-limited environments, is CKD [16–18]. Although highly successful antiretroviral medication exists, the prevalence of CKD in PLWH is rising [19]. The pathogenesis of CKD in PLWH involves several factors, including potentially nephrotoxic ARV drugs and co-medications in addition to cardiovascular disease (CVD), hypertension, diabetes, smoking, as well as chronic systemic inflammation, and HIV viremia [20,21]. Moreover, exposure to TDF may result in proximal renal tubular failure, which is typically asymptomatic but occasionally manifests as Fanconi's syndrome in these individuals [22].

The objectives of this study were to determine the association between having TDF-induced nephrotoxicity at three months and CKD within five years among PLWH

on TDF in Zambia, to determine the incidence, factors associated with CKD, and trend of kidney function in the study population. The age group of interest was adolescents and adults because pediatric populations (0–16) often have distinct physiological and pharmacokinetic profiles, which might require separate investigations.

With 38.4 million PLWH and about 650,000 AIDS-related fatalities reported in 2021 [23], human immunodeficiency virus (HIV) infection continues to be a serious global public health issue [1,24]. Sub-Saharan Africa, where more than two-thirds (70%) of people with HIV now live and where more than half of all HIV-related fatalities occur, bears the brunt of this burden the most [8,25]. With a population of 19.6 million [26],Zambia is a low-to-middle income country (LMIC) that contributes around 21% of Africa's HIV disease burden and over 5% of the continent's AIDS-related mortality [8]. Despite antiretroviral therapy (ART) having significantly enhanced the life expectancy of HIV patients, it also put them at risk for chronic non-communicable diseases such as chronic kidney disease (CKD) [27].

Tenofovir disoproxil fumarate (TDF), the first-line ARV-combination drug used in Zambia has been proven to cause nephrotoxicity among persons living with HIV (PLWH) receiving treatment in different countries across the globe [8]. The drug can damage proximal tubules resulting in AKI [8]. AKI is associated with an increased risk of death and morbidity, including CKD, end-stage kidney disease (ESKD), and other indirect complications, including CVD [8,28,29]. A recognised risk factor for the onset of CKD is recurrent episodes of AKI [16,30] and a lack of chronicity reporting [31]. Consequently, most countries including Zambia are replacing TDF-based regimens with Tenofovir Alafenamide (TAF) which is believed to be safer and less toxic. Therefore, this study is aimed at determining the association between TDF-induced nephrotoxicity and CKD among HIV-Positive Zambian Adults.

Given that TDF is a component of the first-line ART regimen that is administered over an extended length of time and is linked to nephrotoxicity [32] it is important to determine whether TDF-induced nephrotoxicity progresses to chronic kidney disease following three months of treatment initiation and within five years of receiving TDF treatment. Clinicians and other healthcare professionals would benefit from early detection of TDF-induced renal dysfunction because it would help them assess renal function and provide better treatment for HIV/AIDS patients taking TDF-based regimens [33]. This would allow patients to benefit fully from TDF-based regimens. The results of a recent study done in Zambia, which recommended further research in this area, could only predict nephrotoxicity up to the first three months of TDF use. It was pointed out that because there are no widely acknowledged risk models available to identify whether patients are at increased risk for worse kidney outcomes before starting ART, patients in Zambia are treated empirically without assessing the risk for TDF-associated nephrotoxicity [8]. The findings could not tell whether the nephrotoxicity led to CKD later on [8]. The necessity to investigate whether CKD symptoms like proteinuria, tubular dysfunction, or TDF use are connected to a progressive loss in renal function or other negative outcomes is also highlighted by other research investigations in Africa [34]. Therefore, this study presents a picture of what the situation is like in the Zambian setting regarding prolonged treatment with a TDF-based regimen.

This study was trying to answer whether patients who suffer TDF-associated nephrotoxicity at three months proceed to CKD at six months or not and we hypothesized that there is an association between having TDF-associated nephrotoxicity and developing CKD within five years?

Antiretroviral therapy's (ART) main objective is to reduce HIV-related morbidity and mortality [35]. To achieve and maintain a plasma HIV-1 ribonucleic acid (RNA) or viral load below the quantitative limits of commercially available assays, effective ART is used. Prolonged viral suppression [36] lengthens the lives of people with HIV and boosts immune system performance and general quality of life. It also reduces the danger of problems that can lead to AIDS and other conditions [37,38]. By the end of December 2021, 28.7 million of the 38.4 million PLWH in the year 2021 had access to ART [24]. An initial antiretroviral (ARV) regimen for a person with HIV typically consists of two nucleoside reverse transcriptase inhibitors (NRTIs) [39] administered in combination with a third active ARV drug from one of three drug classes [40]: an

integrase strand transfer inhibitor (INSTI), a non-nucleoside reverse transcriptase inhibitor (NNRTI), or a protease inhibitor (PI) with a pharmacokinetic enhancer [37,41]. For adults and adolescents (including women and adolescent girls who are of childbearing potential or are pregnant) the preferred first-line ART regimen consists of TDF and lamivudine (3TC) or emtricitabine (FTC) plus dolutegravir (DTG), and alternative regimen consists of TDF and 3TC plus efavirenz (EFV) 400 mgb [42]. Two NRTIs and either lopinavir/ritonavir (LPV/r) or atazanavir/ritonavir (ATV/r) are the recommended second-line ART. The third line consists of DTG, one to two NRTIs, and darunavir/ritonavir (DRV/r) (if possible, consider optimization using genotyping) [42,43]

Despite the widespread use of ART, the prevalence of CKD is growing, and it is increasingly linked to antiretroviral toxicity and common non-infectious comorbidities (NICMs) [16]. There are significant differences that are readily apparent, with the African continent having the greatest prevalence of CKD among PLWH [16]. Several studies on renal dysfunction among PLWH on TDF have been conducted across the globe and conflicting findings reported [44]. According to a systematic review and meta-analysis on the incidence of CKD worldwide among HIV-positive persons, the prevalence of CKD was 12.3% with Cockcroft-Gault (C-G), 4.8% with CKD Epidemiology Collaboration (CKD-EPI), and 6.4% with Modification of Diet in Renal Disease (MDRD) [16,45]. There were 61 eligible publications consisting of a total of 209,078 PLWH chosen from 60 different countries [16,45]. Regional CKD prevalence was reported to be 3.7% for Europe, 5.7% for Asia, 7.1% for North America, and 7.9% for Africa [45,46]. It was reported that Africa had the greatest MDRD-based prevalence of 7.9%, according to a subgroup analysis of prevalence by WHO region [16]. The pooled MDRD-based prevalence in Africa ranged from 3.2% in Southern Africa to 14.6% in West Africa [16,45]. The researchers' findings were summarized in their conclusion, which noted that CKD is common in HIV-infected individuals, particularly in Africa [16,45], and that HIV treatment programs need to increase their CKD screening efforts in addition to introducing international standards for the diagnosis and treatment of CKD in HIV positive patients [45].

A cohort study among United States veterans with HIV reported odds of TDF-associated CKD of about 48% [47]. The study comprised 4,630 patients with HIV who were exposed to TDF and 1,181 who were never exposed [48]. The conclusion was that Veterans with HIV who are currently exposed to TDF have a higher risk of developing CKD and osteoporotic fractures than the general population as a whole within the VA healthcare system [48]. In 2021, a study was published that comprised a total number of 9802 participants of which 6222 initiated TDF and 3580 did not [6]. The findings of this study were that 125 participants developed CKD over 24 382 person-years of follow-up. In this sizable cohort of PLWH without prior ART experience, it was concluded that incident CKD after ART initiation was uncommon and significantly correlated with pre-existing CKD risk [6]. In those with a low baseline CKD risk, the majority of ART-naive PLWH, and TDF-containing regimens did not increase the risk of CKD, and they may still be an effective treatment choice in the right circumstances [6]. A study conducted in Thailand, among 27,313 participants, showed that 245 patients (representing 0.9%) developed CKD [49]. The study's findings confirmed the current advice that patients with renal function impairment should have their doses adjusted [49]. The risk of CKD was generally low, concentrated largely in those getting TDF [49].

Findings from a 12-year observational cohort study among Asians on ART, with low body weight, indicated a high incidence of CKD, and TDF use was reported to be significantly associated with the development of CKD [50]. Furthermore, it was concluded that after the advent of ART, the rate of estimated eGFR decline was rapid, especially when TDF was used, and that the findings call for ongoing, frequent monitoring of renal function in Asian patients with HIV-1 infection [50]. The researchers suggested additional extensive research to prove the long-term renal prognosis of Asian patients with HIV-1 infection. In Korea, a study was conducted on 210 PLWH, among them were 108 participants on TDF-based ART and 102 were abacavir (ABC)-based ART [51]. Renal dysfunction was reported in 16 patients representing 14.8% in the TDF group and 11 representing 10.8% in the ABC group [52]. It was concluded that approximately, 13% of PLWH treated with TDF had renal dysfunction and that the advanced stage of HIV infection was a significant risk factor for renal dysfunction [51]. It was determined that HIV patients on antiretroviral therapy, primarily the highly active antiretroviral therapy (HAART), have a relatively high prevalence of CKD of 15.3% and that a high level of sCr was predictive of CKD in

the study's logistic regression model [19]. This cross-sectional study was carried out in Nigeria in 2018 on a population of 118 PLWH [19]. The combination of TDF/3TC/EFV and tenofovir-based combination ARV therapy in this study was said to have had a significant impact on CKD in HIV patients [19].

In Ghana, a total of 330 PLWH were assessed. 101 of them were taking TDF and were reported to have developed renal injury characterised by proteinuria and tubular dysfunction (TD) [34]. This was said to be a common adverse effect among PLWH on TDF in Ghana [34]. In the conclusion, it was suggested that further longitudinal studies to determine whether proteinuria, TD, or TDF use are linked to progressive decline in renal function or other adverse outcomes are needed in Africa [34]. A cross-sectional analytical study conducted in Ivory Coast reported a CKD prevalence of 10.4% on a cohort population comprising 402 PLWH and the conclusion was that data are urgently needed to help further characterize the severity of CKD burden in the HIV community [46]. Therefore, being aware of the prevalence of CKD and the risk factors for developing it aids in its detection and can support the clinical judgment of healthcare workers [46]. In Ethiopia, a hospital-based Cross-Sectional Study conducted in 2019, on 243 participants reported a prevalence of CKD (eGFR < 60 ml/min/1.73 m2) at 4.53% [53]. The researchers concluded that CKD in PLWH receiving a TDF-based regimen requires attention and that it was significantly associated with Age > 50 years old, and having cancer as comorbidities [53]. They further pointed out the need for regular monitoring of patients for early diagnosis and management of CKD in a TDF-based regimen [53]. With a CKD prevalence of 1.4% (from a total of 1993 participants) as reported in the findings of the study conducted in Namibia, it was concluded that the findings point to a low prevalence of CKD in northern Namibia's HIV-positive community [54]. One of the studies conducted in South Africa, comprising 489 participants, reported a CKD prevalence ranging from 6.4% to 8.7%, and in their conclusion, researchers pointed out the lack of chronicity reporting among other issues [31].

Most studies conducted in Zambia on PLWH generally looked at renal dysfunction not specifically TDF-induced CKD. One study (comprising 68,628 participants) conducted in Zambia reported an overall prevalence of renal impairment of 6.9% and concluded that a sizeable fraction of PLHIV in Zambia starting ART had moderate to severe kidney function impairment using standard serum creatinine measurements [55]. 2.5% of people over the age of 50 had severe renal function impairment or kidney failure [55]. Unfortunately, the study could not specifically associate TDF use with CKD. Another study comprising 205 participants, conducted at the University Teaching Hospital only reported on TDF-induced nephrotoxicity of 22%, after three months [8].

HIV multiplies and spreads throughout the body using the mechanisms of CD4 cells [56]. This process is referred to as the HIV replication cycle [57]. It is completed in seven steps or stages namely, attachment, fusion, reverse transcription, integration, replication, assembly, and budding [58] as shown in Fig 1.

ART is targeted at these steps as the sites of action [59]. HIV first attaches to a particular location on the surface of a CD4 cell known as a CD4 receptor before starting to enter the cell [60]. The Cysteine-Cysteine Chemokine Receptor Type 5 (CCR5) [61] co-receptor or the C-X-C Chemokine Receptor 4 (CXCR4) [62] co-receptor is the second receptor that HIV must connect to [63]. As a result, the virus can fuse or join with the CD4 cell [64]. HIV distributes its genetic information and enzymes, which are proteins that trigger chemical reactions, into the CD4 cell after fusion. This step is targeted by a class of ARVs called fusion inhibitors (FIs) [65]. HIV is a member of the family Retroviridae and the species Lentivirus [66]. Therefore, its genetic material is RNA which encodes "instructions" that will rewire the CD4 cell to increase viral production. HIV's RNA must be converted into double-stranded deoxyribonucleic acid (dsDNA) to work [67]. Reverse transcriptase, an HIV enzyme, converts HIV RNA into HIV DNA [68]. With ART, this step is blocked by NRTIs and NNRTIs [69].

The newly generated HIV dsDNA then moves into the CD4 cell's nucleus [70], or "command center". Integrase is a different HIV enzyme that joins or integrates HIV's dsDNA with the DNA of CD4 cells [71,72] and this step is the site of action for INSTI class of ARVs. Following its integration into the genome of the CD4 cell, the virus instructs the CD4 cell to begin producing new HIV proteins [73]. The building elements of new HIV viruses are these proteins [74]. They are made in extensive chains and broken down into tiny fragments by a protease enzyme [74]. A new virus is created as the smaller

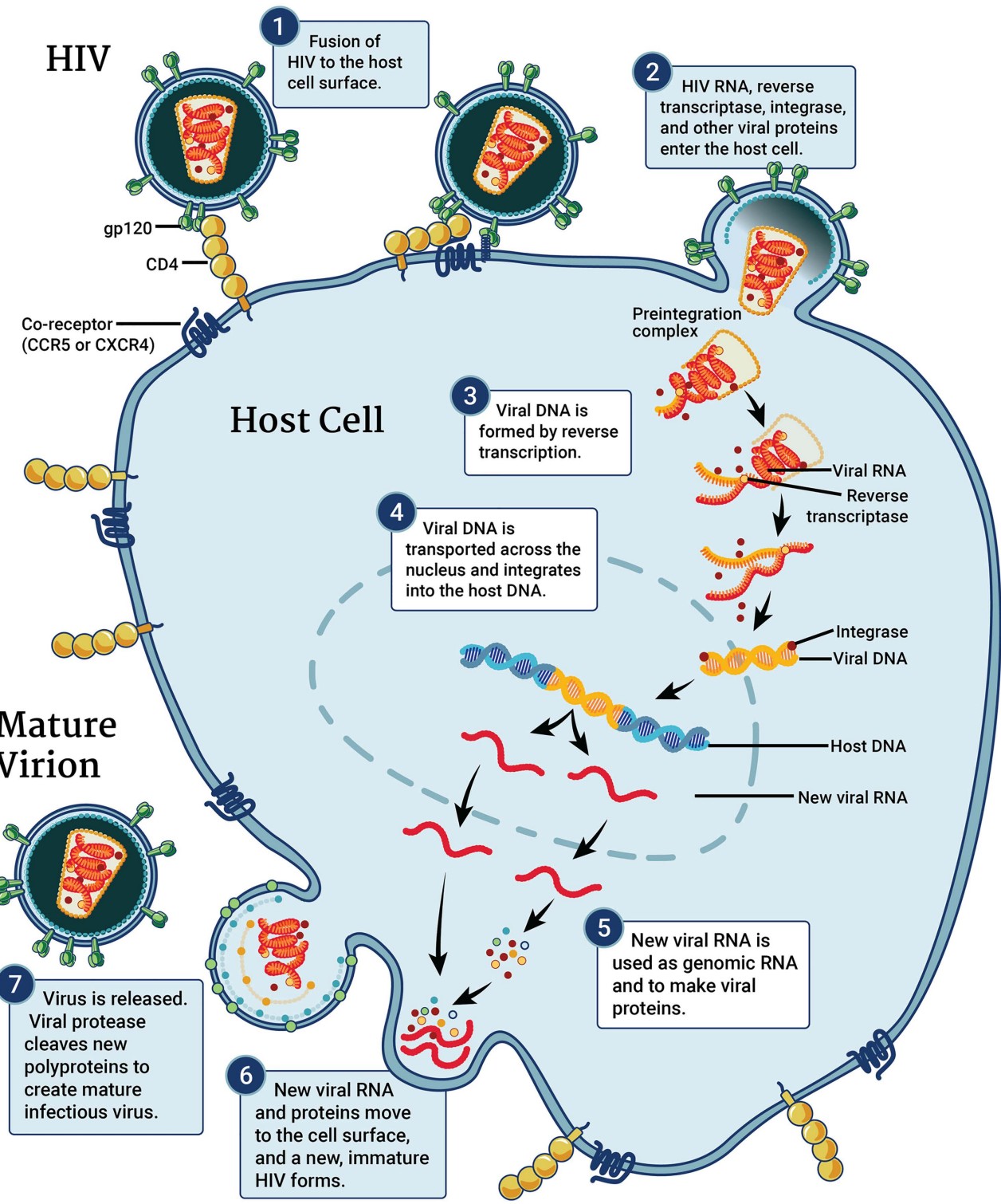

**Fig 1. HIV replication cycle.**

protein fragments and copies of the HIV RNA come together (assembled) [75]. The newly formed virus exit from the original CD4 cell membrane through budding [76]. Other CD4 cells can now be targeted and infected by this novel virus.

Most of the side effects related to ART are more frequently observed in conditions involving chronic use than in sudden start [77]. These include bone perturbations, cardio-metabolic, endocrine, and renal injury [78]. Generally, common and serious adverse effects associated with ART can be grouped according to the ART classes as adapted from the European AIDS Clinical Society Guidelines Version 9.0 October 2017 guidelines [79,80].

NRTIs' common adverse effects include steatosis, peripheral neuropathy, lipoatrophy, and dyslipidemia [79,80]. Server adverse effects associated with drugs in this class of ARVs include ischemic heart disease, systemic hypersensitivity syndrome, rhabdomyolysis, hyperlactatemia, pancreatitis, increased fracture risk, and Fanconi syndrome [79,80]. ARVs in the class of NNRTI have been associated with common adverse effects such as depression, sleep disturbances, headache, dyslipidemia, and lower plasma 25 (OH) vitamin D [79,81]. This is in addition to other severe adverse effects which include suicidal ideation, systemic hypersensitivity, and body rash [79,80]. In the class of PIs, ARVs have been reported to cause dry skin, nausea, diarrhea, hyperbilirubinemia, nephrolithiasis, an increase in abdominal fat, and dyslipidemia [82]. These are grouped under common adverse effects. Severe adverse effects associated with ARVs belonging to the PI class include drug-induced hepatitis, ischemic heart disease, intracranial hemorrhage, and dyslipidemia among others [79,82]. Severe adverse effects such as rhabdomyolysis and systemic hypersensitivity syndrome have been reported and associated with drugs belonging to the class of INSTs [79,80]. Other commonly noted adverse effects associated with this class include nausea, lowering of eGFR, sleep disturbance, and headache [79]. No known common adverse effects have been reported in association with ARVs belonging to the class of FIs but injection nodules have been reported as adverse effects [79].

Changes in bone density have been reported following ART with drugs belonging to NRTIs, NNRTIs, PIs, and INSTIs [83]. ART with TDF has been linked to a larger loss of bone mineral density (BMD) than other NRTIs [84]. Tenofovir Alafenamide (TAF), on the other hand, has been linked to BMD decreases that are less pronounced than those reported with TDF [37]. Bone marrow suppression characterized by anemia, Leukopenia, and neutropenia has been reported or associated with zidovudine (ZDV) [85]. The most common antiretroviral associated with renal disease in first-line antiretroviral treatment is TDF [86]. Gaining weight has been linked to the start of ART [87] and the subsequent viral suppression. When compared to other drug classes, the increase seems to be more pronounced with INSTIs [88]. Moreover, it has been noted that weight gain is higher with TAF than with TDF and doravirine (DOR) with EFV [89]. The use of INSTI has been linked to reports of insomnia, depression, and suicidality, mostly in individuals with pre-existing mental problems [90].

TDF is a nucleotide analog that competes with the natural substrate deoxyadenosine 5′-triphosphate for incorporation into DNA during HIV transcription [40,91]. Tenofovir diphosphate is created by phosphorylating tenofovir, which is hydrolyzed from TDF [92] by gut and plasma esterases. By DNA chain termination and competition with the natural substrate deoxyadenosine 5'-triphosphate [93], tenofovir diphosphate prevents HIV-1 reverse transcriptase from functioning. Glomerular filtration and vigorous proximal tubular secretion help the kidneys get rid of TDF in urine, in an unchanged form [9]. Tenofovir's accumulation and increasing abundance in the proximal tubules would result from its buildup and delayed or impaired excretion [44]. Although the mechanism of TDF nephrotoxicity is not completely understood, it is thought that proximal tubular mitochondrial damage plays a role [94]. Nephrotoxicity caused by tenofovir might show up as Fanconi syndrome [95], AKI, or CKD, and one of its symptoms is proximal tubular cell destruction. The toxic effects of tenofovir are especially dangerous to proximal tubular cells because of their special collection of drug-transporting cell membrane proteins [96]. Tenofovir is thought to work by impairing the activity of mitochondrial DNA-encoded respiratory chain subunits, such as cytochrome c oxidase and NADH dehydrogenase, and by reducing the availability of nucleotides for mitochondrial DNA synthesis, leading to impaired ATP production and mitochondrial injury [97]. TDF buildup in the proximal renal tubular cells can lead to renal toxicity, tubular acidosis, and eventually, renal failure marked by a decrease in eGFR [98]. Some studies have shown that prompt TDF termination (within 6 months of starting treatment) reduces the risk of CKD in individuals who have experienced a moderate drop in eGFR during treatment [10].

More than 800 million people, or more than 10% of the global population, suffer from CKD, a progressive disease [99,100]. It is described as a chronic impairment in kidney structure or function (such as eGFR 60 mL/min/1.73 m2 or albuminuria 30 mg per 24 hours) for more than 3 months [101]. Even if serum creatinine (sCr) has returned to normal, people should be watched for the onset or progression of CKD for at least two to three years after an episode of acute renal injury [102]. Early detection can alter the course of CKD stages 1–3 and prevent complications, but stages 4 and 5 are marked by significant kidney loss, which typically leads to end-stage renal failure [102]. Today, sCr and blood urea levels are typically used to diagnose CKD; however, sCr has been demonstrated to have little predictive value [103]. GFR is used to identify, categorize, and treat CKD, stage the condition, predict mortality and events linked to the condition, and set drug dosages [104].

GFR is calculated using the following equation, which is commonly expressed in terms of volume per time (for example, mL/min): GFR = [UrineX (mg/mL)] * urine flow (mL/min)/ [PlasmaX (mg/mL)], where X is an entirely excreted material [105]. By measuring the creatinine clearance (CrCl), it is possible to determine the approximate GFR as a consequence of the breakdown of dietary meat and the creatinine phosphate found in skeletal muscle [105]. Its body production is reliant on muscular mass [105,106]. Given that the glomerulus freely filters creatinine, the CrCl rate closely resembles the GFR estimate [107,108]. The peritubular capillaries also secrete it, which causes CrCl to overestimate the GFR by 10% to 20% [105,106]. The CrCl (mg/dL) is predicted by the Cockcroft-Gault (C-G) formula using the patient's weight (kg) and gender [109]. If the patient is a female, the resulting CrCl is increased by 0.85 to account for the lower CrCl in females [110]. The primary predictor of CrCl in the C-G formula is age. The formula is as follows: eCCr is calculated as follows: (140 − Age) x Mass (kg) x [0.85 if female]/ 72 x [sCr (mg/dL)] [111]. Alternative formulae for calculating GFR and the variables they utilize to predict GFR include those from the CKD-EPI [112]. The CKD-EPI formulae are divided into groups based on the patient's race, including non-black females, non-black males, and black females [105]. To more accurately calculate GFR in patients with maintained renal function, the Mayo Quadratic formula was created [113]. The Schwartz formula, which takes into account the child's height and sCr (mg/dL), is used to estimate GFR in children (cm) [105] For the estimation of GFR in contemporary clinical practice, the use of creatinine obtained from the Kidney Disease Improving Global Outcomes (KDIGO) clinical practice recommendations is advised [114].

## Methodology

This was a retrospective cohort study on 182 files for adult PLWH conducted at the University Teaching Hospital (UTH) at the Adult Infectious Disease Center (AIDC) in Lusaka, Zambia. These files were for patients confirmed to have Tenofovir-induced Nephrotoxicity after three months of initiating ART. Patients with documented hypertension, diabetes mellitus, and other serious comorbidities known to affect kidney function were excluded during file screening. This step was taken to minimize confounding and focus analysis on nephrotoxicity attributable to TDF exposure. Data was accessed and collected in 01st November 2023 and finished on 31st December 2023.

### Data collection

Demographics, serum creatinine levels with dates, and eGFR (calculated via CKD-EPI 2021 equation) were extracted from medical records over five years. The collected files contained data from January 2018 to December 2023. Data collection from files was done from 1st November 2023 and finished in 31st December 2023. CKD was defined as eGFR < 60 mL/min/1.73m$^2$ for >3 months.

### Statistical analysis

- Python 3.9.0 and R 4.4.3 were used for data analysis

- Incidence Rate was calculated as CKD cases per 1000 person-years.

- Mixed-Effects Models were used to assess longitudinal eGFR trends.

• Cox Regression was used to identify CKD risk factors (sex, baseline creatinine).

### Ethics approval and consent

This study was reviewed and approved by the Levy Mwanawasa Medical University Ethics Committee (Ref: No. 0000141/23) and the National Health Research Authority (NHRA) also granted permission to conduct the study (Ref No: NHRA0002/26/10/2023).

The requirement for informed consent (written, verbal, or otherwise) was waived by the ethics committee and this study did not involve minors. This decision aligns with the retrospective nature of the study and the use of fully anonymized medical records. No personal identifiers were accessible during data analysis. All patient data were fully anonymized prior to researcher access, ensuring confidentiality and compliance with ethical standards.

## Results

### The incidence of CKD among PLWH that initiated TDF in UTH

The 2021 CKD-EPI equation was used to assess kidney function for each of the 182 files of PLWH with TDF-associated nephrotoxicity, and the incidence rate of CKD over a 5-year follow-up period was calculated. A total of 63 participants files (34.62%) developed CKD, contributing to a total person-time of 910 person-years. The incidence rate of CKD was 69.2 cases per 1000 person-years.

Table 1 summarises the univariate analysis statistics for age and baseline eGFR, stratified by gender. Values include frequency, mean, median, standard deviations, minimum, maximum, and range.

Table 2 compares the 5-year CKD incidence in a cohort of 182 HIV patients with TDF-associated nephrotoxicity (34.6% observed incidence) to an external reference population of TDF users without nephrotoxicity (10.4% incidence). Results demonstrate a statistically significant 3.3-fold higher CKD risk (absolute difference: 24.2%, p<0.00001) in the nephrotoxicity group, with a 95% confidence interval (27.7–41.5%) excluding the reference rate.

### Trend of kidney function within five years of developing TDF-associated nephrotoxicity

The plotted trend in Fig 2 below shows repeated eGFR results over five years, measured for patients treated with TDF. This visualization helps understand how kidney function, indicated by eGFR, changes within five years following treatment with TDF.

Table 3 below summarizes the fixed effects (overall trend) and the random effects (variability between patients) from the mixed-effects model analysis.

### Factors associated with CKD and the trend of kidney function in the study population

Table 4 shows the results of the Cox regression model. The first column shows the variable name, the second column shows the coefficient, the third column shows the exponentiated coefficient, the fourth column shows the standard error, the fifth column shows the z-value and the sixth column shows the p-value. The exponentiated coefficient is the hazard

**Table 1. Univariate analysis statistics for age and baseline eGFR.**

| | | Frequency | Mean | Median | Std. Deviation | Minimum | Maximum | Range |
|---|---|---|---|---|---|---|---|---|
| **Age** | female | 107 | 36.87 | 34.86 | 11.59 | 20 | 73 | 53 |
| | male | 75 | 41.08 | 39.8 | 13.04 | 24.6 | 86.58 | 61.98 |
| **Baseline eGFR** | female | 107 | 112.04 | 118.67 | 28.28 | 16.24 | 191.68 | 175.44 |
| | male | 75 | 109.99 | 115.66 | 22.73 | 49.48 | 147.88 | 98.4 |

**Table 2. Comparison of 5-Year CKD Incidence in Patients with TDF-Associated Nephrotoxicity vs. External Reference Population.**

| Metric | Value | Interpretation |
|---|---|---|
| **Observed CKD cases** | 63 | Number of patients who developed CKD after 5 years. |
| **Total cohort size** | 182 | Total number of patients with TDF-associated nephrotoxicity. |
| **Observed proportion (pobs$p_{obs}$)** | 34.6% (0.346) | The proportion of patients who developed CKD. |
| **Reference proportion (p0$p_0$)** | 10.4% (0.104) | 5-year CKD incidence rate in TDF users without nephrotoxicity (external reference). |
| **Z-score** | 10.71 | The test statistic for one-sample Z-test. |
| **p-value** | < 0.00001 | Extremely significant; rejects the null hypothesis. |
| **95% Confidence Interval (CI)** | [27.7%, 41.5%] | The range within which the true CKD rate likely lies. |
| **Absolute risk difference** | 24.2% (0.242) | Excess CKD risk in the cohort compared to the reference. |
| **Relative risk (RR)** | 3.33 | 3.3-fold higher risk of CKD in the cohort compared to the reference. |

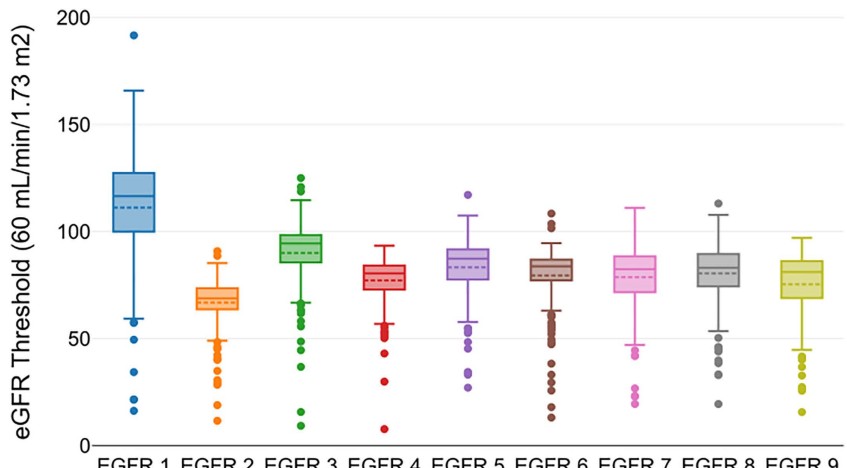

Repeated eGFR Results Over 5 Years

**Fig 2. The plotted trend for repeated eGFR results over 5 years.**

**Table 3. Summary of the mixed-effects model analysis.**

| Model | Intercept Coef. | Intercept Std.Err. | Time Coef. | Time Std.Err. | Group Var | Log-Likelihood | Converged |
|---|---|---|---|---|---|---|---|
| **Basic Model** | 92.729031 | 1.1354233 | −2.045745 | 0.1537114 | 258.00908 | −7006.875 | TRUE |
| **Model without Outliers** | 84.370951 | 1.5407001 | −1.223510 | 0.1619548 | 158.94994 | −3707.589 | TRUE |

**Table 4. Cox proportional hazards regression results.**

| Variables | Coefficients | Exp_Coefficient | Std_Error | Z_Value | P_Value |
|---|---|---|---|---|---|
| **Sex (Male)** | 4.635960 | 103.1266894 | 2.3664568 | 1.9590294 | 0.0502093 |
| **Baseline Creatinine** | 039226 | 1.040005792 | 0.0143650 | 2.7306738 | 0.0063204 |
| **Concordance** | 0.905 (se = 0.069) | | | | |
| **Likelihood ratio Test** | 26.73 on 2 df, p = 2e-06 | | | | |
| **Wald Test** | 12.85 on 2 df, p = 0.002 | | | | |
| **Score (logrank) Test** | 34.14 on 2 df, p = 4e-08 | | | | |

ratio, which is the ratio of the hazard of the event in the group with the variable value of 1 to the hazard of the event in the group with the variable value of 0. For example, the hazard ratio for SEX (male) is 103.13, which means that males have a 103.13 times higher hazard of the event than females. The p-value is the probability of observing the data if the null hypothesis is true, which is that the variable does not affect the hazard of the event. For example, the p-value for sex (male) is 0.0501, which means that there is a 5.01% chance of observing the data if males do not affect the hazard of the event.

Fig 3 illustrates the survival probability across age groups for males and females, based on the Cox Proportional Hazards Model. Notably, fmale exhibit consistently higher survival probabilities than males throughout the age spectrum.

## Discussion

A total of 182 files of PLWH were studied between 2018 and 2023 with a total of 63 (34.62%) files showing to have developed CKD as shown in Table 2, contributing to a total person-time of 910 person-years. The incidence rate of CKD was 69.2 cases per 1000 person-years. This answered the first specific objective of the study.

The mixed-effects model is justified in this study because it effectively accounted for population-level trends and individual variability in CKD progression over time while handling repeated measures and missing data to ensure robust and reliable results. As shown in the results in Table 3 the Intercept Coefficient (Baseline Effect) for the Basic Model (92.73 mL/min/1.73 m$^2$) gave an estimate of kidney function at baseline (time = 0) suggesting a high initial level of kidney function across the cohort. The Model without Outliers showed a decreased intercept (84.37 mL/min/1.73 m$^2$) indicating a lower estimated baseline which suggests that outliers (likely with high baseline kidney function values) significantly influenced the intercept in the basic model. The Time Coefficient (Trend Over Time) in the Basic Model showed a negative time coefficient (−2.05 mL/min/1.73 m$^2$) meaning kidney function decreases over time. This indicates a steeper decline in kidney function per unit of time (e.g., per year). In the Model without Outliers, a less negative time coefficient (−1.22 mL/min/1.73

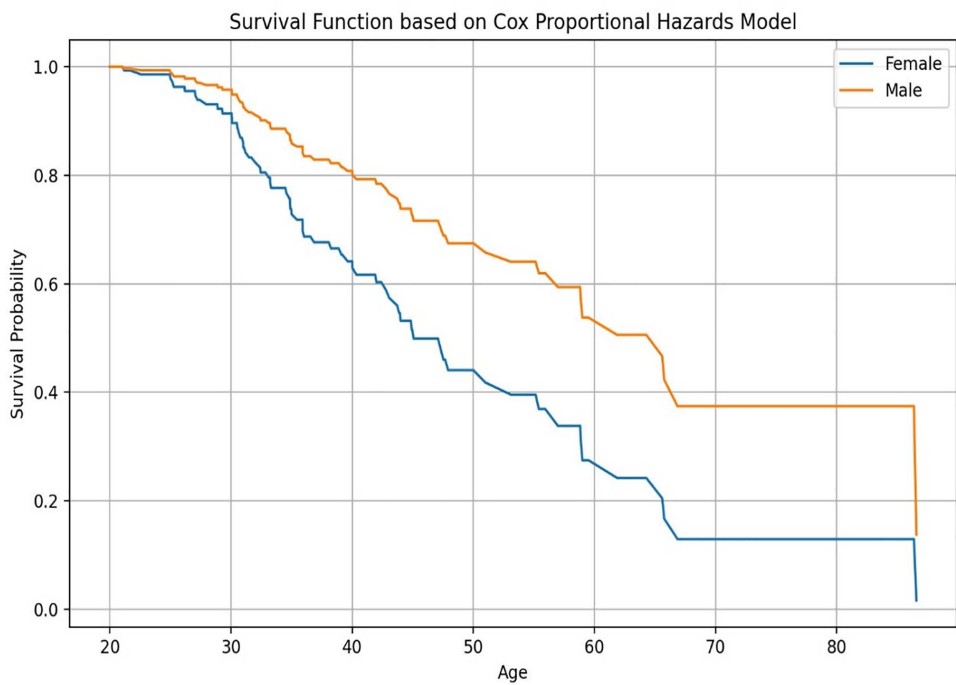

**Fig 3. The plotted trend from a Cox Proportional Hazards model.**

m²) suggests a slower decline in kidney function when outliers are excluded. This indicates that the outliers contributed to a more pronounced decrease in the basic model. The differences in intercepts and time coefficients between the models showed that outliers significantly affect the model's estimates. Removing outliers makes the model more representative of the typical trends in the data, reducing the influence of extreme values. Both models "converged," which means the optimization algorithm used to fit the mixed-effects model successfully found a solution. This is essential for the validity of the results and answers the second specific objective was achieved.

The results of the Cox regression model as shown in Table 4 show that male sex is associated with a higher risk of CKD. The low p-value for this coefficient indicates that this is statistically significant at the 0.05 level. This suggests a statistically significant association between sex and the hazard of the event. These findings are further illustrated in Fig 3, which demonstrates the increased hazard associated with male sex, with survival curves showing a steeper decline compared to females. The hazard ratio for baseline creatinine obtained shows that for every one-unit increase in baseline creatinine, the hazard of the event increases by 4%. The low p-value for the coefficient of this variable shows that this is statistically significant at the 0.01 level suggesting a statistically significant association between baseline creatinine and the hazard of the event. The concordance statistic is a measure of the model's ability to discriminate between individuals with different hazards. The concordance statistic for this model indicates that the model has good discriminatory power. Tests such as the likelihood ratio test, Wald test, and score test were used to assess the overall significance of the model. All three tests were statistically significant, which suggests that the model is a good fit for the data. This information can be used to identify individuals who are at higher risk for CKD and to develop strategies for preventing or delaying the progression of the disease. With the results from the Cox Proportional Hazards Regression, the third specific objective of this study was achieved.

The findings suggest that time is a critical predictor of CKD progression in patients treated with TDF and that a statistically significant decline in eGFR over time highlights the need for regular monitoring and potential interventions to slow kidney function decline. The steepness of the decline (as reflected by the coefficient for time) might vary depending on other covariates like age, baseline eGFR, comorbidities, or medication adherence.

In the discussion of our findings, it is important to consider similar studies that have explored the association between TDF and CKD in PLWH. Our study's observation that TDF exposure may increase the risk of CKD aligns with several international studies. For instance, a cohort study among United States veterans with HIV highlighted increased odds of TDF-associated CKD [47] suggesting that veterans exposed to TDF had a higher risk of developing CKD compared to the general population within the healthcare system [48]. Similarly, a research study conducted in Thailand within a large cohort reported patients to have developed CKD, with a significant risk concentration in those on TDF-based treatment [49]. A 12-year observational cohort study among Asians on antiretroviral therapy (ART) reported a high incidence of CKD, with TDF use being significantly associated with CKD development [50]. In Korea, a study on PLWH noted renal dysfunction within the TDF group (Lee *et al*., 2019b), concluding that PLWH treated with TDF exhibited renal dysfunction, particularly in advanced stages of HIV infection [51]. Countries in Africa that reported similar findings from similar research studied this one include Nigeria [19], Ghana [34], Ethiopia [53], Namibia [54] and South Africa [31]. These studies collectively underscore the importance of vigilant renal monitoring in PLWH who are treated with TDF, reinforcing the findings of our current research in Zambia.

Contradicting findings reported in some studies where TDF was not significantly associated with CKD, were either due to the focus of the research being on other outcomes or the results indicating stable renal function among TDF users. One study reported that people over the age of 50 have had severe renal function impairment or kidney failure [52] but could not specifically associate TDF use with CKD.

The external validity of our findings may be limited by the use of a reference rate ($p_0 = 10.4\%$) derived from a hepatitis B population, rather than HIV-positive patients. CKD risk factors, including HIV-specific systemic inflammation and duration of antiretroviral therapy, may differ significantly between these groups. Additionally, the retrospective design precludes

causal inference regarding the progression from TDF-associated nephrotoxicity to CKD and introduces the possibility of residual confounding. Key clinical variables such as viral load, CD4 count, proteinuria, herbal medication use, smoking status, alcohol intake, and haematocrit were inconsistently documented across patient records and thus excluded from analysis. Notably, hypertension and diabetes mellitus both known CKD risk factors were exclusion criteria in this study, a decision made to reduce confounding and isolate the renal impact of TDF.

## Conclusion and recommendation

The study analysed 182 files of PLWH between 2018 and 2023, with 34.62% developing CKD, leading to an incidence rate of 69.2 cases per 1000 person-years. The mixed-effects model accounted for population-level trends and individual variability, ensuring robust results. Findings showed that removing outliers made the model more representative, with a significant decline in kidney function over time. The Cox regression model identified male sex and baseline creatinine as substantial risk factors for CKD, with good model fit and discriminatory power. Given the high frequency of CKD and the statistically significant deterioration in kidney function over time, it is highly likely that TDF-associated nephrotoxicity plays a major role in the course of CKD in this population. This not only clarifies the study's hypothesis but also shows that baseline renal function and demographic variables like sex have an impact on CKD risk in addition to time.

Although our findings suggest a strong association between TDF exposure and the development of CKD in PLWH, we recognise the limitations inherent to our design, particularly the use of hepatitis B patients as controls and the exclusion of patient files with a history of hypertension and diabetes. As all individuals with early TDF nephrotoxicity were already switched to alternative ART regimens, our results reflect long-term renal health following TDF discontinuation. We recommend ongoing monitoring of kidney function in HIV-positive patients, particularly males and those with elevated baseline creatinine, and advocate for future studies with HIV-only control populations to validate these observations.

## Supporting information

**S1 Data. This dataset includes individual-level laboratory results collected at multiple time points, reflecting renal function trends in participants exposed to tenofovir.** Variables include patient ID (anonymized), date of measurement, serum creatinine (μmol/L), and eGFR (mL/min/1.73m²), calculated using the CKD-EPI formula. The data support the analysis of chronic kidney disease progression and tenofovir-associated nephrotoxicity presented in the main manuscript.
(XLSX)

## Acknowledgments

We thank Levy Mwanawasa Medical University for their support with critical feedback on the research. We also acknowledge the University Teaching Hospital (UTH) Management for granting us access to their data and allowing us to conduct our research studies within their facilities.

The authors are grateful to the anonymous reviewers for their insightful comments, which strengthened the final manuscript.

## Author contributions

**Conceptualization:** Edgar Muchinta, Freeman W. Chabala.

**Data curation:** Edgar Muchinta, Freeman W. Chabala.

**Formal analysis:** Edgar Muchinta, Freeman W. Chabala.

**Investigation:** Edgar Muchinta, Freeman W. Chabala.

**Methodology:** Edgar Muchinta, Freeman W. Chabala.

**Project administration:** Edgar Muchinta.

**Resources:** Edgar Muchinta, Freeman W. Chabala.

**Software:** Freeman W. Chabala.

**Supervision:** Freeman W. Chabala.

**Validation:** Freeman W. Chabala.

**Visualization:** Freeman W. Chabala.

**Writing – original draft:** Edgar Muchinta.

**Writing – review & editing:** Edgar Muchinta, Freeman W. Chabala.

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
