## [Decision Letter · Decision Letter 0]

18 Jul 2025

Dear Dr. Muchinta,

Thank you for submitting your manuscript to PLOS ONE. After careful consideration, we feel that it has merit but does not fully meet PLOS ONE’s publication criteria as it currently stands. Therefore, we invite you to submit a revised version of the manuscript that addresses the points raised during the review process.

We look forward to receiving your revised manuscript.

Kind regards,

Abdulwasiu Bolaji Tiamiyu, MBBS, FMCP

Academic Editor

PLOS ONE

Journal Requirements:

“Non”

Reviewers' comments:

Reviewer's Responses to Questions

**Comments to the Author**

1. Is the manuscript technically sound, and do the data support the conclusions?

Reviewer #1: Partly

Reviewer #2: Yes

2. Has the statistical analysis been performed appropriately and rigorously?

Reviewer #1: No

Reviewer #2: Yes

3. Have the authors made all data underlying the findings in their manuscript fully available?

Reviewer #1: Yes

Reviewer #2: Yes

4. Is the manuscript presented in an intelligible fashion and written in standard English?

Reviewer #1: Yes

Reviewer #2: Yes

Reviewer #1: 1.. Although your experiment was conducted rigorously and appropriate data obtained, some of your conclusions were based on comparison with a study in patients with Hepatitis B. The study will be more robust and the conclusion more validated if you had used HIV patients without tenofovir toxicity as your controls. As it is now, your conclusion that HIV patients with Tenofovir toxicity have an increased risk of CKD can not be validated

2. There is no description of patients’ baseline characteristics apart from age, gender and baseline creatinine. What about other clinicodemographic characteristics such as smoking status, level of alcohol intake, consumption of herbal medications, BMI, CD4 count levels, viral load, proteinuria level, Haematocrit and presence of other comorbidities like hypertension and Diabetes Mellitus. Adding this to your result section will make your result more robust, provide a better platform for comparison with other studies and allow for determining if your result is applicable to other patient populations. It would also allow you to perform more univariate analyses to determine factors associated with development of CKD in patients with tenofovir toxicity.

Reviewer #2: The manuscript is technically sound and the data provided supported the claims and findings. The data is also made available. The language of the manuscript is appropriate for scholastic communication. The statitical analysis was performed appropriately and scientifically sound. The study aimed to determine the association between TDF-induced nephrotoxicity at three months and chronic kidney disease (CKD) within five years among people living with HIV (PLWH) in Zambia. The findings indicated a significant association between TDF-associated nephrotoxicity and the

development of CKD within five years. This claim was properly placed in contex of previous literature as the author conducted extensive literature review. The data analyses fully supported the claims. The study protocol was meticulously explained. The study conformed to relevant guidelines. The methodology was comprehensive enough to allow other researchers to replicate it in other centers. The language of the manuscript is clear enough to be accessible to those who are not specialists in the field.

**Do you want your identity to be public for this peer review?** For information about this choice, including consent withdrawal, please see our Privacy Policy

Reviewer #1: **Yes: ** Dr Habibu Aliyu Galadanci

Reviewer #2: **Yes: ** DR MOJEED OLAITAN RAFIU

---

## [Author Response · Author response to Decision Letter 1]

23 Jul 2025

Reviewer #1 Comments and Author Responses

1. Concern: Use of external comparison with Hepatitis B patients for CKD incidence rates.

Response: We appreciate this observation. The comparison with Hepatitis B patients was derived from a previously published study and served only as contextual framing for our incidence rate in Table 2. We have now clarified this in the revised Discussion section. Furthermore, the Limitations section now acknowledges the difference in pathophysiology and treatment exposures between Hepatitis B and HIV populations and notes that future studies should incorporate HIV-specific control cohorts.

2. Concern: Missing clinicodemographic characteristics such as CD4 count, viral load, smoking status, alcohol intake, proteinuria, and comorbidities.

Response: Thank you for this valuable recommendation. We have addressed this by:

• Explicitly stating in the revised Methodology that hypertension, diabetes mellitus, and other chronic comorbidities were part of our exclusion criteria, and this step was taken to reduce confounding.

• Expanding the Limitations section to explain that CD4 count, viral load, proteinuria, smoking status, alcohol use, herbal medication intake, and haematocrit were not consistently recorded due to the retrospective nature of the study and therefore excluded from analysis. We noted that future prospective studies could overcome this gap to enable deeper univariate analyses.

Reviewer #2 Comments and Author Response

We thank Reviewer #2 for their encouraging feedback and endorsement of our study’s technical rigor, statistical methodology, and scholarly communication. We were pleased to see that the data were deemed appropriately analyzed and the manuscript clearly written and methodologically sound.

Editorial Requirements Response

1. Abstract Consistency: We have ensured that the abstract in the manuscript and the online submission form are now fully aligned.

2. Competing Interests Statement: We updated this section to: “The authors have declared that no competing interests exist.”

3. Data Availability Statement: We included the statement: “All data are in the manuscript and/or supporting information files.” placed under a dedicated Data Availability heading.

4. Formatting Compliance: We reviewed and updated the manuscript to comply with PLOS ONE formatting guidelines, including file naming conventions and reference layout.

5. Figure Standards: All figure files have been verified to comply with the journal's figure submission criteria using PACE.

We trust these revisions fully address the concerns raised and strengthen the manuscript for publication. We remain grateful to the reviewers for their constructive comments and to the editorial team for their guidance and professionalism.

---

## [Editor Report · Decision Letter 1]

31 Jul 2025

Chronic Kidney Disease among HIV-positive Zambian adults with Tenofovir-associated Nephrotoxicity at University Teaching Hospital (UTH) in Lusaka.

PONE-D-25-19997R1

Dear Dr. Muchinta,

We’re pleased to inform you that your manuscript has been judged scientifically suitable for publication and will be formally accepted for publication once it meets all outstanding technical requirements.

Kind regards,

Abdulwasiu Bolaji Tiamiyu, MBBS, FMCP

Academic Editor

PLOS ONE
---

## [Editor Report · Acceptance letter]

PONE-D-25-19997R1

PLOS ONE

Dear Dr. Muchinta,

I'm pleased to inform you that your manuscript has been deemed suitable for publication in PLOS ONE. Congratulations! Your manuscript is now being handed over to our production team.

Kind regards,

on behalf of

Dr. Abdulwasiu Bolaji Tiamiyu

Academic Editor

PLOS ONE